# New Liftable Classes for
# First-Order Probabilistic Inference

**Seyed Mehran Kazemi**
The University of British Columbia
smkazemi@cs.ubc.ca

**Angelika Kimmig**
KU Leuven
angelika.kimmig@cs.kuleuven.be

**Guy Van den Broeck**
University of California, Los Angeles
guyvdb@cs.ucla.edu

**David Poole**
The University of British Columbia
poole@cs.ubc.ca

## Abstract

Statistical relational models provide compact encodings of probabilistic dependencies in relational domains, but result in highly intractable graphical models. The goal of lifted inference is to carry out probabilistic inference without needing to reason about each individual separately, by instead treating exchangeable, undistinguished objects as a whole. In this paper, we study the *domain recursion* inference rule, which, despite its central role in early theoretical results on domain-lifted inference, has later been believed redundant. We show that this rule is more powerful than expected, and in fact significantly extends the range of models for which lifted inference runs in time polynomial in the number of individuals in the domain. This includes an open problem called S4, the symmetric transitivity model, and a first-order logic encoding of the birthday paradox. We further identify new classes $S^2FO^2$ and $S^2RU$ of domain-liftable theories, which respectively subsume $FO^2$ and *recursively unary theories*, the largest classes of domain-liftable theories known so far, and show that using domain recursion can achieve exponential speedup even in theories that cannot fully be lifted with the existing set of inference rules.

## 1  Introduction

Statistical relational learning (SRL) [8] aims at unifying logic and probability for reasoning and learning in noisy domains, described in terms of individuals (or objects), and the relationships between them. Statistical relational models [10], or template-based models [18] extend Bayesian and Markov networks with individuals and relations, and compactly describe probabilistic dependencies among them. These models encode exchangeability among the objects: individuals that we have the same information about are treated similarly.

A key challenge with SRL models is the fact that they represent highly intractable, densely connected graphical models, typically with millions of random variables. The aim of *lifted inference* [23] is to carry out probabilistic inference without needing to reason about each individual separately, by instead treating exchangeable, undistinguished objects as a whole. Over the past decade, a large number of lifted inference rules have been proposed [5, 9, 11, 14, 20, 22, 28, 30], often providing exponential speedups for specific SRL models. These basic exact inference techniques have applications in (tractable) lifted learning [32], where the main task is to efficiently compute *partition functions*, and in variational and over-symmetric approximations [29, 33]. Moreover, they provided the foundation for a rich literature on approximate lifted inference and learning [1, 4, 13, 17, 19, 21, 25, 34].

The theoretical study of lifted inference began with the complexity notion of *domain-lifted* inference [31] (a concept similar to data complexity in databases). Inference is domain-lifted when it runs in time polynomial in the number of individuals in the domain. By identifying *liftable classes* of models, guaranteeing domain-lifted inference, one can characterize the theoretical power of the various inference rules. For example, the class $FO^2$, encoding dependencies among pairs of individuals (i.e., two logical variables), is liftable [30]. Kazemi and Poole [15] introduce a liftable class called *recursively unary*, capturing hierarchical simplification rules. Beame *et al.* [3] identify liftable classes of probabilistic database queries. Such results elevate the specific inference rules and examples to a general principle, and bring lifted inference in line with complexity and database theory [3].

This paper studies the *domain recursion* inference rule, which applies the principle of induction on the domain size. The rule makes one individual $A$ in the domain explicit. Afterwards, the other inference rules simplify the SRL model up to the point where it becomes identical to the original model, except the domain size has decreased. Domain recursion was introduced by Van den Broeck [31] and was central to the proof that $FO^2$ is liftable. However, later work showed that simpler rules suffice to capture $FO^2$ [27], and the domain recursion rule was forgotten.

We show that domain recursion is more powerful than expected, and can lift models that are otherwise not amenable to domain-lifted inference. This includes an open problem by Beame *et al.* [3], asking for an inference rule for a logical sentence called S4. It also includes the symmetric transitivity model, and an encoding of the birthday paradox in first-order logic. There previously did not exist any efficient algorithm to compute the partition function of these SRL models, and we obtain exponential speedups. Next, we prove that domain recursion supports its own large *classes of liftable models* $S^2FO^2$ subsuming $FO^2$, and $S^2RU$ subsuming recursive unary[1]. All existing exact lifted inference algorithms (e.g., [11, 15, 28]) resort to grounding the theories in $S^2FO^2$ or $S^2RU$ that are not in $FO^2$ or recursively unary, and require time exponential in the domain size.

These results will be established using the weighted first-order model counting (WFOMC) formulation of SRL models [28]. WFOMC is close to classical first-order logic, and it can encode many other SRL models, including Markov logic [24], parfactor graphs [23], some probabilistic programs [7], relational Bayesian networks [12], and probabilistic databases [26]. It is a basic specification language that simplifies the development of lifted inference algorithms [3, 11, 28].

## 2 Background and Notation

A **population** is a set of constants denoting individuals (or objects). A **logical variable (LV)** is typed with a population. We represent LVs with lower-case letters, constants with upper-case letters, the population associated with a LV $x$ with $\Delta_x$, and its cardinality with $|\Delta_x|$. That is, a population $\Delta_x$ is a set of constants $\{X_1, \ldots, X_n\}$, and we use $x \in \Delta_x$ as a shorthand for instantiating $x$ with one of the $X_i$. A **parametrized random variable (PRV)** is of the form $\mathsf{F}(t_1, \ldots, t_k)$ where $\mathsf{F}$ is a predicate symbol and each $t_i$ is a LV or a constant. A **unary** PRV contains exactly one LV and a **binary** PRV contains exactly two LVs. A **grounding** of a PRV is obtained by replacing each of its LVs $x$ by one of the individuals in $\Delta_x$.

A **literal** is a PRV or its negation. A **formula** $\varphi$ is a literal, a disjunction $\varphi_1 \vee \varphi_2$ of formulas, a conjunction $\varphi_1 \wedge \varphi_2$ of formulas, or a quantified formula $\forall x \in \Delta_x : \varphi(x)$ or $\exists x \in \Delta_x : \varphi(x)$ where $x$ appears in $\varphi(x)$. A **sentence** is a formula with all LVs quantified. A **clause** is a disjunction of literals. A **theory** is a set of sentences. A theory is **clausal** if all its sentences are clauses. An **interpretation** is an assignment of values to all ground PRVs in a theory. An interpretation $I$ is a **model** of a theory $T$, $I \models T$, if given its value assignments, all sentences in $T$ evaluate to True.

Let $\mathcal{F}(T)$ be the set of predicate symbols in theory $T$, and $\Phi : \mathcal{F}(T) \to \mathbb{R}$ and $\overline{\Phi} : \mathcal{F}(T) \to \mathbb{R}$ be two functions that map each predicate $\mathsf{F}$ to weights. These functions associate a weight with assigning True or False to ground PRVs $\mathsf{F}(C_1, \ldots, C_k)$. For an interpretation $I$ of $T$, let $\psi^{True}$ be the set of ground PRVs assigned True, and $\psi^{False}$ the ones assigned False. The weight of $I$ is given by $\omega(I) = \prod_{\mathsf{F}(C_1,\ldots,C_k) \in \psi^{True}} \Phi(\mathsf{F}) \cdot \prod_{\mathsf{F}(C_1,\ldots,C_k) \in \psi^{False}} \overline{\Phi}(\mathsf{F})$. Given a theory $T$ and two functions $\Phi$ and $\overline{\Phi}$, the **weighted first-order model count (WFOMC)** of the theory given $\Phi$ and $\overline{\Phi}$ is: $\mathrm{WFOMC}(T|\Phi, \overline{\Phi}) = \sum_{I \models T} \omega(I)$.

In this paper, we assume that all theories are clausal and do not contain existential quantifiers. The latter can be achieved using the Skolemization procedure of Van den Broeck *et al.* [30], which efficiently transforms a theory $T$ with existential quantifiers into a theory $T'$ without existential quantifiers that has the same weighted model count. That is, our theories are sets of finite-domain, function-free first-order clauses whose LVs are all universally quantified (and typed with a population). Furthermore, when a clause mentions two LVs $x_1$ and $x_2$ with the same population $\Delta_x$, or a LV $x$ with population $\Delta_x$ and a constant $C \in \Delta_x$, we assume they refer to different individuals.[2]

**Example 1.** Consider the theory $\forall x \in \Delta_x : \neg\mathsf{Smokes}(x) \vee \mathsf{Cancer}(x)$ having only one clause and assume $\Delta_x = \{A, B\}$. The assignment $\mathsf{Smokes}(A) = \mathsf{True}, \mathsf{Smokes}(B) = \mathsf{False}, \mathsf{Cancer}(A) = \mathsf{True}, \mathsf{Cancer}(B) = \mathsf{True}$ is a model. Assuming $\Phi(\mathsf{Smokes}) = 0.2, \Phi(\mathsf{Cancer}) = 0.8, \overline{\Phi}(\mathsf{Smokes}) = 0.5$ and $\overline{\Phi}(\mathsf{Cancer}) = 1.2$, the weight of this model is $0.2 \cdot 0.5 \cdot 0.8 \cdot 0.8$. This theory has eight other models. The WFOMC can be calculated by summing the weights of all nine models.

## 2.1 Converting Inference for SRL Models into WFOMC

For many SRL models, (lifted) inference can be converted into a WFOMC problem. As an example, consider a Markov logic network (MLN) [24] with weighted formulae $(w_1 : F_1, \ldots, w_k : F_k)$. For every weighted formula $w_i : F_i$ of this MLN, let theory $T$ have a sentence $\mathsf{Aux}_i(x, \ldots) \Leftrightarrow F_i$ such that $\mathsf{Aux}_i$ is a predicate having all LVs appearing in $F_i$. Assuming $\Phi(\mathsf{Aux}_i) = \exp(w_i)$, and $\Phi$ and $\overline{\Phi}$ are 1 for the other predicates, the *partition function* of the MLN is equal to WFOMC$(T)$.

## 2.2 Calculating the WFOMC of a Theory

We now describe a set of rules $\mathcal{R}$ that can be applied to a theory to find its WFOMC efficiently; for more details, readers are directed to [28], [22] or [11]. We use the following theory $T$ with two clauses and four PRVs ($\mathsf{S}(x, m)$, $\mathsf{R}(x, m)$, $\mathsf{T}(x)$ and $\mathsf{Q}(x)$) as our running example:

$$\forall x \in \Delta_x, m \in \Delta_m : \mathsf{Q}(x) \vee \mathsf{R}(x, m) \vee \mathsf{S}(x, m) \qquad \forall x \in \Delta_x, m \in \Delta_m : \mathsf{S}(x, m) \vee \mathsf{T}(x)$$

**Lifted Decomposition** Assume we ground $x$ in $T$. Then the clauses mentioning an arbitrary $X_i \in \Delta_x$ are $\forall m \in \Delta_m : \mathsf{Q}(X_i) \vee \mathsf{R}(X_i, m) \vee \mathsf{S}(X_i, m)$ and $\forall m \in \Delta_m : \mathsf{S}(X_i, m) \vee \mathsf{T}(X_i)$. These clauses are totally disconnected from clauses mentioning $X_j \in \Delta_x$ ($j \neq i$), and are the same up to renaming $X_i$ to $X_j$. Given the exchangeability of the individuals, we can calculate the WFOMC of only the clauses mentioning $X_i$ and raise the result to the power of the number of connected components ($|\Delta_x|$). Assuming $T_1$ is the theory that results from substituting $x$ with $X_i$, WFOMC$(T) = $ WFOMC$(T_1)^{|\Delta_x|}$.

**Case-Analysis** The WFOMC of $T_1$ can be computed by a case analysis over different assignments of values to a ground PRV, e.g., $\mathsf{Q}(X_i)$. Let $T_2$ and $T_3$ represent $T_1 \wedge \mathsf{Q}(X_i)$ and $T_1 \wedge \neg\mathsf{Q}(X_i)$ respectively. Then, WFOMC$(T_1) = $ WFOMC$(T_2) + $ WFOMC$(T_3)$. We follow the process for $T_3$ (the process for $T_2$ will be similar) having clauses $\neg\mathsf{Q}(X_i), \forall m \in \Delta_m : \mathsf{Q}(X_i) \vee \mathsf{R}(X_i, m) \vee \mathsf{S}(X_i, m)$ and $\forall m \in \Delta_m : \mathsf{S}(X_i, m) \vee \mathsf{T}(X_i)$.

**Unit Propagation** When a clause in the theory has only one literal, we can propagate the effect of this clause through the theory and remove it[3]. In $T_3$, $\neg\mathsf{Q}(X_i)$ is a unit clause. Having this unit clause, we can simplify the second clause and get the theory $T_4$ having clauses $\forall m \in \Delta_m : \mathsf{R}(X_i, m) \vee \mathsf{S}(X_i, m)$ and $\forall m \in \Delta_m : \mathsf{S}(X_i, m) \vee \mathsf{T}(X_i)$.

**Lifted Case-Analysis** Case-analysis can be done for PRVs having one logical variable in a lifted way. Consider the $\mathsf{S}(X_i, m)$ in $T_4$. Due to the exchangeability of the individuals, we do not have to consider all possible assignments to all ground PRVs of $\mathsf{S}(X_i, m)$, but only the ones where the number of individuals $M \in \Delta_m$ for which $\mathsf{S}(X_i, M)$ is $\mathsf{True}$ (or equivalently $\mathsf{False}$) is different. This means considering $|\Delta_m| + 1$ cases suffices, corresponding to $\mathsf{S}(X_i, M)$ being $\mathsf{True}$ for exactly $j = 0, \ldots, |\Delta_m|$ individuals. Note that we must multiply by $\binom{|\Delta_m|}{j}$ to account for the number

of ways one can select $j$ out of $|\Delta_m|$ individuals. Let $T_{4j}$ represent $T_4$ with two more clauses: $\forall m \in \Delta_{m_T} : \mathsf{S}(X_i, m)$ and $\forall m \in \Delta_{m_F} : \neg\mathsf{S}(X_i, m)$, where $\Delta_{m_T}$ represents the $j$ individuals in $\Delta_m$ for which $\mathsf{S}(X_i, M)$ is True, and $\Delta_{m_F}$ represents the other $|\Delta_m| - j$ individuals. Then $\mathrm{WFOMC}(T_4) = \sum_{j=0}^{|\Delta_m|} \binom{|\Delta_m|}{j} \mathrm{WFOMC}(T_{4j})$.

**Shattering**   In $T_{4j}$, the individuals in $\Delta_m$ are no longer exchangeable: we know different things about those in $\Delta_{m_T}$ and those in $\Delta_{m_F}$. We need to shatter every clause having individuals coming from $\Delta_m$ to make the theory exchangeable. To do so, the clause $\forall m \in \Delta_m : \mathsf{R}(X_i, m) \vee \mathsf{S}(X_i, m)$ in $T_{4j}$ must be shattered to $\forall m \in \Delta_{m_T} : \mathsf{R}(X_i, m) \vee \mathsf{S}(X_i, m)$ and $\forall m \in \Delta_{m_F} : \mathsf{R}(X_i, m) \vee \mathsf{S}(X_i, m)$ (and similarly for the other formulae). The shattered theory $T_{5j}$ after unit propagation will have clauses $\forall m \in \Delta_{m_F} : \mathsf{R}(X_i, m)$ and $\forall m \in \Delta_{m_F} : \mathsf{T}(X_i)$.

**Decomposition, Caching, and Grounding**   In $T_{5j}$, the two clauses have different PRVs, i.e., they are disconnected. In such cases, we apply decomposition, i.e., find the WFOMC of each connected component separately and return the product. The WFOMC of the theory can be found by continuing to apply the above rules. In all the above steps, after finding the WFOMC of each (sub-)theory, we store the results in a cache so we can reuse them if the same WFOMC is required again. By following these steps, one can find the WFOMC of many theories in polynomial time. However, if we reach a point where none of the above rules are applicable, we ground one of the populations which makes the process exponential in the number of individuals.

### 2.3   Domain-Liftability

The following notions allow us to study the power of a set of lifted inference rules.

**Definition 1.**  A theory is **domain-liftable** [31] if calculating its WFOMC is polynomial in $|\Delta_{x_1}|, |\Delta_{x_2}|, \ldots, |\Delta_{x_k}|$ where $x_1, x_2, \ldots, x_k$ represent the LVs in the theory. A class $C$ of theories is domain-liftable if $\forall T \in C$, $T$ is domain-liftable.

So far, two main classes of domain-liftable theories have been recognized: $FO^2$ [30, 31] and *recursively unary* [15, 22].

**Definition 2.**  A theory is in $FO^2$ if all its clauses have up to two LVs.

**Definition 3.**  A theory $T$ is *recursively unary (RU)* if for every theory $T'$ resulting from applying rules in $\mathcal{R}$ except for lifted case analysis to $T$, until no more rules apply, there exists some unary PRV in $T'$ and a generic case of lifted case-analysis on this unary PRV is itself *RU*.

Note that the time needed to check whether a theory is in $FO^2$ or $RU$ is independent of the domain sizes in the theory. For $FO^2$, the membership check can be done in time linear in the size of the theory, whereas for *RU*, only a worst-case exponential procedure is known. Thus, $FO^2$ currently offers a faster membership check than *RU*, but as we show later, *RU* subsumes $FO^2$. This gives rise to a trade-off between fast membership checking and modeling power for, e.g., lifted learning purposes.

## 3   The Domain Recursion Rule

Van den Broeck [31] considered another rule called *domain recursion* in the set of rules for calculating the WFOMC of a theory. The intuition behind domain recursion is that it modifies a domain $\Delta_x$ by making one element explicit: $\Delta_x = \Delta_{x'} \cup \{A\}$ with $A \notin \Delta_{x'}$. Next, clauses are rewritten in terms of $\Delta_{x'}$ and $A$ while removing $\Delta_x$ from the theory entirely. Then, by applying standard rules in $\mathcal{R}$ on this modified theory, the problem is reduced to a WFOMC problem on a theory identical to the original one, except that $\Delta_x$ is replaced by the smaller domain $\Delta_{x'}$. This lets us compute WFOMC using dynamic programming. We refer to $\mathcal{R}$ extended with the domain recursion rule as $\mathcal{R}^{\mathcal{D}}$.

**Example 2.**  Suppose we have a theory whose only clause is $\forall x, y \in \Delta_p : \neg\mathsf{Friend}(x, y) \vee \mathsf{Friend}(y, x)$, stating if $x$ is friends with $y$, $y$ is also friends with $x$. One way to calculate the WFOMC of this theory is by grounding only one individual in $\Delta_p$ and then using $\mathcal{R}$. Let $A$ be an individual in $\Delta_p$ and let $\Delta_{p'} = \Delta_p - \{A\}$. We can (using domain recursion) rewrite the theory as: $\forall x \in \Delta_{p'} : \neg\mathsf{Friend}(x, A) \vee \mathsf{Friend}(A, x)$, $\forall y \in \Delta_{p'} : \neg\mathsf{Friend}(A, y) \vee \mathsf{Friend}(y, A)$, and $\forall x, y \in \Delta_{p'} : \neg\mathsf{Friend}(x, y) \vee \mathsf{Friend}(y, x)$. Lifted case-analysis on $\mathsf{Friend}(p', A)$ and $\mathsf{Friend}(A, p')$,

shattering and unit propagation give $\forall x, y \in \Delta_{p'} : \neg\mathsf{Friend}(x, y) \vee \mathsf{Friend}(y, x)$. This theory is equivalent to our initial theory, with the only difference being that the population of people has decreased by one. By keeping a cache of the values of each sub-theory, one can verify that this process finds the WFOMC of the above theory in polynomial time.

Note that the theory in Example 2 is in $FO^2$ and as proved in [27], its WFOMC can be computed without using the domain recursion rule[4]. This proof has caused the domain recursion rule to be forgotten in the lifted inference community. In the next section, we revive this rule and identify a class of theories that are only domain-liftable when using the domain recursion rule.

## 4 Domain Recursion Makes More Theories Domain-Liftable

In this section, we show three example theories that are not domain-liftable when using $\mathcal{R}$, yet become domain-liftable with domain recursion.

**S4 Clause:** Beame *et al.* [3] identified a clause (S4) with four binary PRVs having the same predicate and proved that, even though the rules $\mathcal{R}$ in Section 2.2 cannot calculate the WFOMC of that clause, there is a polynomial-time algorithm for finding its WFOMC. They concluded that this set of rules $\mathcal{R}$ for finding the WFOMC of theories does not suffice, asking for new rules to compute their theory. We prove that adding domain recursion to the set achieves this goal.

**Proposition 1.** *The theory consisting of the S4 clause* $\forall x_1, x_2 \in \Delta_x, y_1, y_2 \in \Delta_y : \mathsf{S}(x_1, y_1) \vee \neg\mathsf{S}(x_2, y_1) \vee \mathsf{S}(x_2, y_2) \vee \neg\mathsf{S}(x_1, y_2)$ *is domain-liftable using* $\mathcal{R}^{\mathcal{D}}$.

**Symmetric Transitivity:** Domain-liftable calculation of WFOMC for the transitivity formula is a long-standing open problem. Symmetric transitivity is easier as its model count corresponds to the Bell number, but solving it using general-purpose rules has been an open problem. Consider clauses $\forall x, y, z \in \Delta_p : \neg\mathsf{F}(x, y) \vee \neg\mathsf{F}(y, z) \vee \mathsf{F}(x, z)$ and $\forall x, y \in \Delta_p : \neg\mathsf{F}(x, y) \vee \mathsf{F}(y, x)$ defining a symmetric transitivity relation. For example, $\Delta_p$ may indicate the population of people and $\mathsf{F}$ may indicate friendship.

**Proposition 2.** *The symmetric-transitivity theory is domain-liftable using* $\mathcal{R}^{\mathcal{D}}$.

**Birthday Paradox:** The birthday paradox problem [2] is to compute the probability that in a set of $n$ randomly chosen people, two of them have the same birthday. A first-order encoding of this problem requires computing the WFOMC for a theory with clauses $\forall p \in \Delta_p, \exists d \in \Delta_d : \mathsf{Born}(p, d)$, $\forall p \in \Delta_p, d_1, d_2 \in \Delta_d : \neg\mathsf{Born}(p, d_1) \vee \neg\mathsf{Born}(p, d_2)$, and $\forall p_1, p_2 \in \Delta_p, d \in \Delta_d : \neg\mathsf{Born}(p_1, d) \vee \neg\mathsf{Born}(p_2, d)$, where $\Delta_p$ and $\Delta_d$ represent the population of people and days. The first two clauses impose the condition that every person is born in exactly one day, and the third clause states the "no two people are born on the same day" query.

**Proposition 3.** *The birthday-paradox theory is domain-liftable using* $\mathcal{R}^{\mathcal{D}}$.

## 5 New Domain-Liftable Classes: $S^2FO^2$ and $S^2RU$

In this section, we identify new domain-liftable classes, enabled by the domain recursion rule.

**Definition 4.** Let $\alpha(\mathsf{S})$ be a clausal theory that uses a single binary predicate $\mathsf{S}$, such that each clause has exactly two different literals of $\mathsf{S}$. Let $\alpha = \alpha(\mathsf{S}_1) \wedge \alpha(\mathsf{S}_2) \wedge \cdots \wedge \alpha(\mathsf{S}_n)$ where the $\mathsf{S}_i$ are different binary predicates. Let $\beta$ be a theory where all clauses contain at most one $\mathsf{S}_i$ literal, and the clauses that contain an $\mathsf{S}_i$ literal contain no other literals with more than one LV. Then, $S^2FO^2$ and $S^2RU$ are the classes of theories of the form $\alpha \wedge \beta$ where $\beta \in FO^2$ and $\beta \in RU$ respectively.

**Theorem 1.** $S^2FO^2$ *and* $S^2RU$ *are domain-liftable using* $\mathcal{R}^{\mathcal{D}}$.

*Proof.* The case where $\alpha = \emptyset$ is trivial. Let $\alpha = \alpha(\mathsf{S}_1) \wedge \alpha(\mathsf{S}_2) \wedge \cdots \wedge \alpha(\mathsf{S}_n)$. Once we remove all PRVs having none or one LV by (lifted) case-analysis, the remaining clauses can be divided into $n + 1$ components: the $i$-th component in the first $n$ components only contains $\mathsf{S}_i$ literals, and the

$(n + 1)$-th component contains no $S_i$ literals. These components are disconnected from each other, so we can consider each of them separately. The $(n + 1)$-th component comes from clauses in $\beta$ and is domain-liftable by definition. The following two Lemmas prove that the clauses in the other components are also domain-liftable. The proofs of both lemmas rely on domain recursion.

**Lemma 1.** *A clausal theory $\alpha(\mathsf{S})$ with only one predicate $\mathsf{S}$ where all clauses have exactly two different literals of $\mathsf{S}$ is domain-liftable.*

**Lemma 2.** *Suppose $\{\Delta_{p_1}, \Delta_{p_2}, \ldots, \Delta_{p_n}\}$ are mutually exclusive subsets of $\Delta_x$ and $\{\Delta_{q_1}, \Delta_{q_2}, \ldots, \Delta_{q_m}\}$ are mutually exclusive subsets of $\Delta_y$. We can add any unit clause of the form $\forall p_i \in \Delta_{p_i}, q_j \in \Delta_{q_j} : \mathsf{S}(p_i, q_j)$ or $\forall p_i \in \Delta_{p_i}, q_j \in \Delta_{q_j} : \neg\mathsf{S}(p_i, q_j)$ to the theory $\alpha(\mathsf{S})$ in Lemma 1 and the theory is still domain-liftable.*

Therefore, theories in $S^2 FO^2$ and $S^2 RU$ are domain-liftable. $\hfill\square$

It can be easily verified that membership checking for $S^2 FO^2$ and $S^2 RU$ is not harder than for $FO^2$ and $RU$, respectively.

**Example 3.** Suppose we have a set $\Delta_j$ of jobs and a set $\Delta_v$ of volunteers. Every volunteer must be assigned to at most one job, and every job requires no more than one person. If the job involves working with gas, the assigned volunteer must be a non-smoker. And we know that smokers are most probably friends with each other. Then we will have the following first-order theory:

$$\forall v_1, v_2 \in \Delta_v, j \in \Delta_j : \neg\mathsf{Assigned}(v_1, j) \lor \neg\mathsf{Assigned}(v_2, j)$$
$$\forall v \in \Delta_v, j_1, j_2 \in \Delta_j : \neg\mathsf{Assigned}(v, j_1) \lor \neg\mathsf{Assigned}(v, j_2)$$
$$\forall v \in \Delta_v, j \in \Delta_j : \mathsf{InvolvesGas}(j) \land \mathsf{Assigned}(v, j) \Rightarrow \neg\mathsf{Smokes}(v)$$
$$\forall v_1, v_2 \in \Delta_v : \mathsf{Aux}(v_1, v_2) \Leftrightarrow (\mathsf{Smokes}(v_1) \land \mathsf{Friends}(v_1, v_2) \Rightarrow \mathsf{Smokes}(v_2))$$

Predicate Aux is added to capture the probability assigned to the last rule (as in MLNs). This theory is not in $FO^2$, not in $RU$, and is not domain-liftable using $\mathcal{R}$. However, the first two clauses are of the form described in Lemma 1, the third and fourth are in $FO^2$ (and also in $RU$), and the third clause, which contains $\mathsf{Assigned}(v, j)$, has no other PRVs with more than one LV. Therefore, this theory is in $S^2 FO^2$ (and also in $S^2 RU$) and domain-liftable based on Theorem 1.

**Example 4.** Consider the birthday paradox introduced in Section 4. After Skolemization [30] for removing the existential quantifier, the theory contains $\forall p \in \Delta_p, \forall d \in \Delta_d : \mathsf{S}(p) \lor \neg\mathsf{Born}(p, d)$, $\forall p \in \Delta_p, d_1, d_2 \in \Delta_d : \neg\mathsf{Born}(p, d_1) \lor \neg\mathsf{Born}(p, d_2)$, and $\forall p_1, p_2 \in \Delta_p, d \in \Delta_d : \neg\mathsf{Born}(p_1, d) \lor \neg\mathsf{Born}(p_2, d)$, where $\mathsf{S}$ is the Skolem predicate. This theory is not in $FO^2$, not in $RU$, and is not domain-liftable using $\mathcal{R}$. However, the last two clauses belong to clauses in Lemma 1, the first one is in $FO^2$ (and also in $RU$) and has no PRVs with more than one LV other than Born. Therefore, this theory is in $S^2 FO^2$ (and also in $S^2 RU$) and domain-liftable based on Theorem 1.

**Proposition 4.** $FO^2 \subset RU$, $FO^2 \subset S^2 FO^2$, $FO^2 \subset S^2 RU$, $RU \subset S^2 RU$, $S^2 FO^2 \subset S^2 RU$.

*Proof.* Let $T \in FO^2$ and $T'$ be any of the theories resulting from exhaustively applying rules in $\mathcal{R}$ except lifted case analysis on $T$. If $T$ initially contains a unary PRV with predicate $\mathsf{S}$, either it is still unary in $T'$ or lifted decomposition has replaced the LV with a constant. In the first case, we can follow a generic branch of lifted case-analysis on $\mathsf{S}$, and in the second case, either $T'$ is empty or all binary PRVs in $T$ have become unary in $T'$ due to applying the lifted decomposition and we can follow a generic branch of lifted case-analysis for any of these PRVs. The generic branch in both cases is in $FO^2$ and the same procedure can be followed until all theories become empty. If $T$ initially contains only binary PRVs, lifted decomposition applies as the grounding of $T$ is disconnected for each pair of individuals, and after lifted decomposition all PRVs have no LVs. Applying case analysis on all PRVs gives empty theories. Therefore, $T \in RU$. The theory $\forall x, y, z \in \Delta_p : \mathsf{F}(x, y) \lor \mathsf{F}(y, z) \lor \mathsf{F}(x, y, z)$ is an example of a $RU$ theory that is not in $FO^2$, showing $RU \not\subset FO^2$. $FO^2$ and $RU$ are special cases of $S^2 FO^2$ and $S^2 RU$ respectively, where $\alpha = \emptyset$, showing $FO^2 \subset S^2 FO^2$ and $RU \subset S^2 RU$. However, Example 3 is both in $S^2 FO^2$ and $S^2 RU$ but is not in $FO^2$ and not in $RU$, showing $S^2 FO^2 \not\subset FO^2$ and $S^2 RU \not\subset RU$. Since $FO^2 \subset RU$ and the class of added $\alpha(S)$ clauses are the same, $S^2 FO^2 \subset S^2 RU$. $\hfill\square$

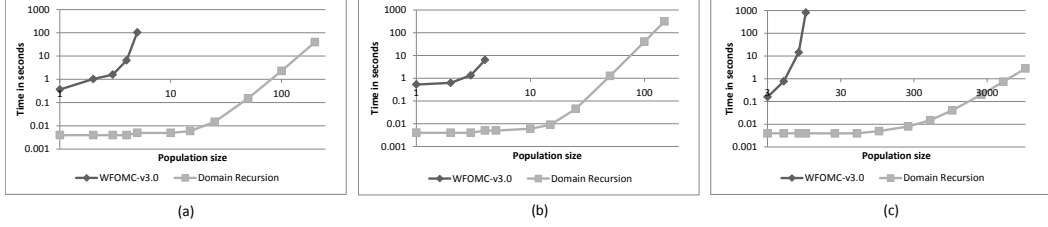

Figure 1: Run-times for calculating the WFOMC of (a) the theory in Example 3, (b) the S4 clause, and (c) symmetric transitivity, using the WFOMC-v3.0 software (which only uses $\mathcal{R}$) and comparing it to the case where we use the domain recursion rule, referred to as *Domain Recursion* in the diagrams.

## 6    Experiments and Results

In order to see the effect of using domain recursion in practice, we find the WFOMC of three theories with and without using the domain recursion rule: (a) the theory in Example 3, (b) the S4 clause, and (c) the symmetric-transitivity theory. We implemented the domain recursion rule in C++ and compiled the codes using the g++ compiler. We compare our results with the WFOMC-v3.0 software[5]. Since this software requires domain-liftable input theories, for the first theory we grounded the jobs, for the second we grounded $\Delta_x$, and for the third we grounded $\Delta_p$. For each of these three theories, assuming $|\Delta_x| = n$ for all LVs $x$ in the theory, we varied $n$ and plotted the run-time as a function of $n$. All experiments were done on a 2.8GH core with 4GB RAM under MacOSX. The run-times are reported in seconds. We allowed a maximum of 1000 seconds for each run.

Obtained results can be viewed in Fig. 1. These results are consistent with our theory and indicate the clear advantage of using the domain recursion rule in practice. In Fig. 1(a), the slope of the diagram for domain recursion is approximately 4 which indicates the degree of the polynomial for the time complexity. Similar analysis can be done for the results on the $S4$ clause and the symmetric-transitivity clauses represented in Fig. 1(b), (c). The slope of the diagram in these two diagrams is around 5 and 2 respectively, indicating that the time complexity for finding their WFOMC are $n^5$ and $n^2$ respectively, where $n$ is the size of the population.

## 7    Discussion

We can categorize theories with respect to the domain recursion rule as: (1) theories proved to be domain-liftable using domain recursion (e.g., S4, symmetric transitivity, and theories in $S^2FO^2$), (2) theories that are domain-liftable using domain recursion, but we have not identified them yet as such, and (3) theories that are not domain-liftable even when using domain recursion. We leave discovering and characterizing the theories in category 2 and 3 as future work. But here we show that even though the theories in category 3 are not domain-liftable using domain recursion, this rule may still result in exponential speedups for these theories.

Consider the (non-symmetric) transitivity rule: $\forall x, y, z \in \Delta_p : \neg\mathsf{Friend}(x, y) \vee \neg\mathsf{Friend}(y, z) \vee \mathsf{Friend}(x, z)$. Since none of the rules in $\mathcal{R}$ apply to the above theory, the existing lifted inference engines ground $\Delta_p$ and calculate the weighted model count (WMC) of the ground theory. By grounding $\Delta_p$, these engines lose great amounts of symmetry. Suppose $\Delta_p = \{A, B, C\}$ and assume we select $\mathsf{Friend}(A, B)$ and $\mathsf{Friend}(A, C)$ as the first two random variables for case-analysis. Due to the exchangeability of the individuals, the case where $\mathsf{Friend}(A, B)$ and $\mathsf{Friend}(A, C)$ are assigned to True and False respectively has the same WMC as the case where they are assigned to False and True. However, the current engines fail to exploit this symmetry as they consider grounded individuals non-exchangeable.

By applying domain recursion to the above theory instead of fully grounding it, one can exploit the symmetries of the theory. Suppose $\Delta_{p'} = \Delta_p - \{P\}$. Then we can rewrite the theory as follows:

$$\forall y, z \in \Delta_{p'} : \neg\mathsf{Friend}(P, y) \vee \neg\mathsf{Friend}(y, z) \vee \mathsf{Friend}(P, z)$$

$$\forall x, z \in \Delta_{p'} : \neg\mathsf{Friend}(x, P) \vee \neg\mathsf{Friend}(P, z) \vee \mathsf{Friend}(x, z)$$
$$\forall x, y \in \Delta_{p'} : \neg\mathsf{Friend}(x, y) \vee \neg\mathsf{Friend}(y, P) \vee \mathsf{Friend}(x, P)$$
$$\forall x, y, z \in \Delta_{p'} : \neg\mathsf{Friend}(x, y) \vee \neg\mathsf{Friend}(y, z) \vee \mathsf{Friend}(x, z)$$

Now if we apply lifted case analysis on $\mathsf{Friend}(P, y)$ (or equivalently on $\mathsf{Friend}(P, z)$), we do not get back the same theory with reduced population and calculating the WFOMC is still exponential. However, we only generate one branch for the case where $\mathsf{Friend}(P, y)$ is True only once. This branch covers both the symmetric cases mentioned above. Exploiting these symmetries reduces the time-complexity exponentially.

This suggests that for any given theory, when the rules in $\mathcal{R}$ are not applicable one may want to try the domain recursion rule before giving up and resorting to grounding a population.

## 8  Conclusion

We identified new classes of domain-liftable theories called $S^2FO^2$ and $S^2RU$ by reviving the domain recursion rule. We also demonstrated how this rule is useful for theories outside these classes. Our work opens up a future research direction for identifying and characterizing larger classes of theories that are domain-liftable using domain recursion. It also helps us get closer to finding a dichotomy between the theories that are domain-liftable and those that are not, similar to the dichotomy result of Dalvi and Suciu [6] for query answering in probabilistic databases.

It has been shown [15, 16] that compiling the WFOMC rules into low-level programs (e.g., C++ programs) offers a (approx.) 175x speedup compared to other approaches. While compiling the previously known rules to low-level programs was straightforward, compiling the domain recursion rule to low-level programs without using recursion might be tricky as it relies on the population size of the logical variables. A future research direction would be finding if the domain recursion rule can be efficiently compiled into low-level programs, and measuring the amount of speedup it offers.

**Acknowledgements.** AK is supported by the Research Foundation Flanders (FWO). GVdB is partially supported by NSF (#IIS-1633857).

## Footnotes

[1]All proofs can be found in the extended version of the paper at: `https://arxiv.org/abs/1610.08445`

[2]Equivalently, we can disjoin $x_1 = x_2$ or $x = C$ to the clause.

[3]Note that unit propagation may remove clauses and random variables from the theory. To account for them, *smoothing* multiplies the WFOMC by $2^{\#rv}$, where $\#rv$ represents the number of removed variables.

[4]This can be done by realizing that the theory is disconnected in the grounding for every pair $(A, B)$ of individuals and applying the lifted case-analysis.

[5]Available at: https://dtai.cs.kuleuven.be/software/wfomc

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
