[Reviews · NeurIPS 2016]

Reviewer 1

Summary

This paper shows that an inference rule called domain recursion improves on state-of-the-art algorithms for lifted probabilistic inference, broadening the set of problems that can be solved in time polynomial in the population size. This rule was introduced in 2011, but apparently it is no longer included in current packages because it was thought to be superseded by other rules. The paper also defines a class of models called FO2+, which includes some of the models that require the domain recursion rule for tractable inference.

Qualitative Assessment

This paper makes a concrete contribution to lifted probabilistic inference, showing that the domain recursion rule can be used to solve certain interesting problems that are intractable for state-of-the-art lifted inference software. The insights here seem likely to be incorporated into upcoming versions of those packages. However, some of the statements in the paper are not sufficiently precise. The description of the domain recursion rule itself (p. 5, top) is much less precise than the definition in the 2011 paper that introduced it. It's not clear what the preconditions are for applying the rule or exactly how it transforms the theory. Also, the description mentions caching (line 175), but it would be helpful to explain how the inference algorithm ends up making multiple calls to the cache. Another key issue is the definition of a "recursively unary" theory (page 4, line 155). This term was introduced in an earlier paper, but I found the definition there to be imprecise as well. What particularly confused me was the use of this definition in the proof of Proposition 4 (page 7, line 259). The proof refers to a theory where "each connected component of the theory mentions the same two LVs [logical variables]". This seems to be exactly the kind of theory that does *not* satisfy the definition of recursively unary, which calls for each component to include a literal with at most one LV. I think what's going on is that each component is being thought of as grounded, which means it has no LVs left. But then why can't we just ground any theory and claim that it's recursively unary? This definition needs to be more precise. I'm actually not convinced that introducing the terms "recursively unary" and "FO2+" is helpful. The class of theories with at most two LVs per clause (FO2) is a very natural class, so it's helpful to know that that class is liftable. But are more baroque definitions of liftable classes really more helpful than just seeing whether one can find a series of applications of the given lifted inference rules to solve a theory without grounding? We need to think about such sequences of operations anyway for problems outside of FO2+, such as S4 and symmetric transitivity. Smaller comments: - p. 6, Lemmas 1 and 2: It would be good to say where the domain recursion rule comes into play here. Which lemma depends on that rule? - p. 7, Figure 1: Should include actual numbers for the population sizes. - p. 7, line 269: "We implemented the domain recursion rule for these three theories". Have you actually integrated this rule into a full lifted inference algorithm, or did you just implement it specially for these theories? == Update After Author Feedback == I appreciate the authors' responses to my comments. Regarding the proof of Proposition 5, I see how lifted decomposition can be applied there. However, it's still unclear to me exactly how that case is covered by the definition of "recursively unary" (RU). I see that the grounding of the theory has a connected component for each pair of individuals. But if we're talking about the grounding of a theory, then it no longer makes sense for the definition of RU to talk about the number of logical variables -- the grounding has no logical variables. Is there a way to precisely define the ground-level connected components of a non-ground theory? The response also says that there are no preconditions for applying domain recursion. But domain recursion can't always be applied *efficiently* (in time polynomial in the domain sizes), can it? What if our theory has several domains on which we could recurse -- is there a way to choose which one to recurse on?

Confidence in this Review

2-Confident (read it all; understood it all reasonably well)


Reviewer 2

Summary

The authors return to "domain-recursion" as a technique for lifted inference (that is, inference, focused on models that combine probabilities and logical construction, that avoids grounding the whole model). The technique is shown to tackle problems that have defied liftable inference so far. Also the authors characterise a new class of liftable models. Small scale tests show the gains from the technique (somewhat unsurprisingly, as we know that liftable inference, whenever applicable, may lead to exponential gains).

Qualitative Assessment

The authors present a novel assessment of domain-recursion, and show that it can handle previously open problems. This is very interesting, and the paper is clear in its main goals and results. Overall the paper is an incremental effort in revisiting previous results and previous algorithms; the contribution is focused and narrow, but it can be quite valuable for other researchers interested in this topic. I should note that this is a difficult topic. The paper requires some polishing but still it contains a contribution. Concerning the presentation, I find that the authors must add some comments on their assumptions. For instance, what exactly is the language they allow (I know that previous papers on this subject follow the same path, but still this is a something that must be fixed). I mean, is a formula an expression without quantifiers, and understood as universally quantified? But then what is a "theory in first-order logic" mentioned in page 2 (is it with quantifiers)? And then why is it that the birthday paradox uses an existential quantifier (and then the authors mention that it can be dealt with by Skolemization, but how? The exact way Skolemization is used should be shown). I find that the description of the language in Section 2 should be clear and leave no space for doubts. The authors must improve that aspect of the paper. Later, the same effort should be made to clarify FO^2 and its variants (do they accept quantifiers? where? how?). Note for instance that if formulas are actually open formulas without quantifiers, then it is hard to imagine how they are interpreted for weighted first-order model count; I guess such a formula must be closed (but then, I suppose all formulas are implicitly closed with universal quantifiers?). A relatively small point: I find Section 2.1 not to be very useful, the example with MLN is interesting but not needed here. Perhaps this can be shortened so that space is left for more important material. A point about notation: it is funny that a PRV, say Q, appears as q in formula, while a logical variable, say x, appear capitalized as X; what is the logic concerning capitalization here? There seems to be something wrong with the expression in the section about Shattering; the first expression has two quantifiers, but it should only have one (??). It is a pity that S4-clauses and symmetric transitivities are not shown, however shortly, to be liftable. They are not in FO^2+, so the reader is left without any clue as to why they are liftable. Perhaps some space can be found to add some explanation on this matter. I find the definition of FO^2+ to be harder to parse than necessary; please improve Definition 1. One point I did not quite get: can one know the gains that will be obtained from domain-recursion BEFORE applying it? I mean, suppose we have a model and we cannot solve by other rules. We check and see that the model is NOT in FO^2+. Can we know whether domain-recursion will be useful, or not? At the end of Section 6, the authors mention that costs are n^5 and n^2. How was that concluded? Just by empirical testing indicated in the graphs? But the gains should be derived from the algorithms that are used; if it is liftable against something exponential, the gains are exponential. I did not get the meaning of n^5 and n^2 there. Concerning the text, here are some specific comments: - Page 1, line -4: the concept of domain-liftability may seem similar to data complexity, but actually lqe-liftability as defined by Van den Broeck and coauthors is more akin to data complexity; I don't think domain complexity has such a clear counterpart in the database world. - Is footnote 1 correct (perhaps conjoin X != Y)? - Right before Section 2.3: "process exponential in the number of individuals" is quite imprecise; this depends on the algorithm used to handle grounding, I suppose. - Right before Section 6, the authors use "suggesting", but I suppose they mean "showing". - Footnote 3 misses an ending period. - Page 8: "straightforward" instead of "straight-forward". - Problems with capitalization in references: prolog, markov.

Confidence in this Review

3-Expert (read the paper in detail, know the area, quite certain of my opinion)


Reviewer 3

Summary

The paper describes a new rule for lifted inference called domain recursion and shows that it can be used for domain-lifted inference in relational theories previously thought unliftable.

Qualitative Assessment

Lifted inference is a powerful mechanism for inference in statistical relational models that exploits symmetries from first-order structure. This paper introduces a new rule that augments previous lifted inference rules. The significance of this rule is that it allows domain-lifted exact inference to be carried out in models where previously it needed to be completely grounded. The writing has sufficient clarity and rigor. However, as it has been argued before, in practice, lifting rules such as the ones described in the paper alone might be insufficient for scalable inference, so its practical utility might be limited. However, I think from a theoretical perspective, the paper has sufficient novelty and depth. It can also hopefully open up some new avenues for approximations.

Confidence in this Review

3-Expert (read the paper in detail, know the area, quite certain of my opinion)


Reviewer 4

Summary

The authors introduce a new class of first-order probabilistic models liftable using the domain-recursion rule. The new class FO2+ expands upon the previous class of liftable models, FO2 by allowing formulae with two literals with the same predicate and no restriction on the variables. They empirically show that their approach scales to larger population sizes by using domain recursion as compared to state-of-the-art approaches.

Qualitative Assessment

The authors do a great job of setting up prior work and defining the formalism. However, the definition of their new liftable class as well as the proof is then limited to less than one page. This is the main section of the paper that presents their contribution and needs much more details than provided. As a result, the proof is hard to follow and I am still not sure how the authors used domain recursion to lift FO2+. Some questions: - How is S4 and symmetric transitivity liftable when they contain more than two literals for the same PRV ? - Where is domain recursion used in the proof ? A more detailed derivation of Example 4 is necessary to understand their proof and lifted inference approach for this class. Also I feel calling domain recursion as a forgotten rule is a bit of an exaggeration. Apart from this basic presentation issue, the related work and experiments are well written. Expanding the set of liftable models is an important technical contribution and would be very valuable.

Confidence in this Review

1-Less confident (might not have understood significant parts)


Reviewer 5

Summary

The paper demonstrates how the existing domain recursion rule can help exand the set of formula considered liftable in the literature. Three formulas are demonstrated which cannot be solved with existing known rules, but only by adding this rule, and the exisiting class of liftable theories RU is expanded to a new class FO^(2+) by using the newly proposed rule.

Qualitative Assessment

While the results on using domain-recursion to solve new formulas like S4 in section 4 are interesting, they need to be further investigated and extended to make for an interesting research paper. The first omission in the paper is lack of mentions of exact time complexity. For e.g. S4 was already known to be solvable, and since this paper proposes an alternative approach, we need to know how their running times compare: are they same or is one faster. My impression is the approach used in the paper(in appendix) would be slower; if that is true does the rule need to be rephrased so we match the time complexity of the method proposed in [3]. Another missing part is that the newly proposed theory FO^(2+) doesn't even include all the tractable cases proposed in the paper. It would be good to have a definition of tractable instances that includes S4 for completion. So, while its intersting to see how domain-recursion can solve some formula, I feel the paper leaves a lot undiscovered, and the results presented don't warrant a complete research paper.

Confidence in this Review

3-Expert (read the paper in detail, know the area, quite certain of my opinion)


Reviewer 6

Summary

1. The paper describes how a rule called "domain recursion", originally introduced by [Van den Broeck 2011], can be used to solve a more weighted counting problems in polynomial time that was previously thought of in [Beame, Van den Broeck 2015]. 2. The overall problem that the paper tries to address is important and at the forefront of many problems in AI. However, somewhat unsatisfying is that the paper makes rather incremental progress: it shows that an existing approach can be applied to more problems, but does not give any completeness results similar to the dichotomy results by [Dalvi and Suciu 2004]. In Section 7, the paper shows that the approach can be applied to more problems, but again fails to explain when and when not. 3. The paper provides little intuition behind the key insights that were required for the results in the paper. And whether the approach can generalize in some ways that would allow to solve the tractability frontier. The new results are listed from page 5 on. The pages before are dedicated to stating results from [Van den Broeck 2011], [Van den Broeck 2012], [Van den Broeck 2014]. The proofs in the appendix are mechanic and with little intuition. 4. The paper seems to miss foundational work in lifted inference. The paper states: "The theoretical study of lifted inference began with the complexity notion of domain-lifted inference [31] (a concept similar to data complexity in databases) ... "Beame et al. [3] identify liftable classes of probabilistic database queries". [3] Paul Beame, Guy Van den Broeck, Eric Gribkoff, and Dan Suciu. Symmetric weighted first-order model counting. In PODS, pages 313–328, 2015. [31] Guy Van den Broeck. On the completeness of first-order knowledge compilation for lifted probabilistic inference. In Advances in Neural Information Processing Systems, pages 1386–1394, 2011. Dalvi and Suciu presented tractable cases for probabilistic inference in 2004 for which they received the 10-year influential paper award in 2014. This influential work seems to predate this paper's cited papers and included many of the ideas that are also described in this paper, just in other nomenclature. Just one example of many: line 107: "Lifted decomposition": seems to correspond to root variables in Dalvi and Suciu's work

Qualitative Assessment

1. Please include more intuition behind the rules and your novel breakthrough. E.g., while I read example 3 a few times, I am still wondering what the actual reason is that is allows efficiency. You can cut on reviewing prior work but rather cite. 2. Include a discussion of on scalable first order inference that predates [Van den Broeck 2011]

Confidence in this Review

2-Confident (read it all; understood it all reasonably well)